# Tunable even- and odd-denominator fractional quantum Hall states in trilayer graphene

Yiwei Chen[1,12], Yan Huang[1,12], Qingxin Li[1,12], Bingbing Tong[2,3], Guangli Kuang[4], Chuanying Xi[4], Kenji Watanabe[5], Takashi Taniguchi[6], Guangtong Liu[2,3,7] ✉, Zheng Zhu[8], Li Lu[2,3,7], Fu-Chun Zhang[8,9,10], Ying-Hai Wu[11] ✉ & Lei Wang[1,10] ✉

Fractional quantum Hall (FQH) states are exotic quantum many-body phases whose elementary charged excitations are anyons obeying fractional braiding statistics. While most FQH states are believed to have Abelian anyons, the Moore–Read type states with even denominators – appearing at half filling of a Landau level (LL) – are predicted to possess non-Abelian excitations with appealing potential in topological quantum computation. These states, however, depend sensitively on the orbital contents of the single-particle LL wavefunctions and the LL mixing. Here we report magnetotransport measurements on Bernal-stacked trilayer graphene, whose multiband structure facilitates interlaced LL mixing, which can be controlled by external magnetic and displacement fields. We observe robust FQH states including even-denominator ones at filling factors $v = -9/2$, $-3/2$, $3/2$ and $9/2$. In addition, we fine-tune the LL mixing and crossings to drive quantum phase transitions of these half-filling states and neighbouring odd-denominator ones, exhibiting related emerging and waning behaviour.

Electrons confined in a two-dimensional system under a perpendicular magnetic field develop quantized energy levels. And fully filling such Landau levels (LLs) one by one gives rise to the integer quantum Hall states[1]. Within one LL, the strong Coulomb interaction dominates over kinetic energy in the highly degenerated LL flatband, and further conduces to the emergence of FQH states at certain fractional fillings $v$[2]. In most cases, the denominators of $v$ are odd integers, which finds

an explanation in that FQH states can be understood effectively as integer quantum Hall states of composite fermions[3]. An important exception to the odd-denominator rule is the 5/2 FQH state found in the second LL of GaAs[4]. Based on extensive experimental and theoretical investigations[5–9], the most probable explanation of this state is the Moore–Read theory[10] and its extensions, which provide us the Pfaffian, anti-Pfaffian, and particle-hole symmetric Pfaffian

[1]National Laboratory of Solid-State Microstructures, School of Physics, Nanjing University, Nanjing 210093, China. [2]Beijing National Laboratory for Condensed Matter Physics and Institute of Physics, Chinese Academy of Sciences, Beijing 100190, China. [3]Hefei National Laboratory, Hefei 230088, China. [4]Anhui Province Key Laboratory of Condensed Matter Physics at Extreme Conditions, High Magnetic Field Laboratory of the Chinese Academy of Science, Hefei 230031, China. [5]Research Center for Electronic and Optical Materials, National Institute for Materials Science, 1-1 Namiki, Tsukuba 305-0044, Japan. [6]Research Center for Materials Nanoarchitectonics, National Institute for Materials Science, 1-1 Namiki, Tsukuba 305-0044, Japan. [7]Songshan Lake Materials Laboratory, Dongguan 523808, China. [8]Kavli Institute of Theoretical Sciences, University of Chinese Academy of Sciences, Beijing 100049, China. [9]CAS Center for Excellence in Topological Quantum Computation, University of Chinese Academy of Sciences, Beijing 100049, China. [10]Collaborative Innovation Center of Advanced Microstructures, Nanjing University, Nanjing 210093, China. [11]School of Physics and Wuhan National High Magnetic Field Center, Huazhong University of Science and Technology, Wuhan 430074, China. [12]These authors contributed equally: Yiwei Chen, Yan Huang, Qingxin Li. ✉e-mail: gtliu@iphy.ac.cn; yinghaiwu88@hust.edu.cn; leiwang@nju.edu.cn

wavefunctions as candidates[11–14]. The elementary charged excitations of these states obey non-Abelian braiding statistics and may be utilized to perform fault-tolerant quantum computation that are topologically protected at the fundamental level[15]. In recent years, even-denominator FQH states have also been observed in several other systems and some of them are believed to host non-Abelian anyons[16–23].

We build high quality Bernal stacked trilayer graphene (TLG) devices here and report even-denominator FQH states, to our knowledge, for the first time in this system, as well as a plethora of odd-denominator ones. Extensive investigations have been recently carried out on monolayer and bilayer graphene (MLG, BLG)[17–21], and experimental attempts have also been made on TLG. However, FQH states in TLG remained elusive with only tenuous traces of odd-denominator ones speculated[24]. Compared to MLG and BLG, TLG possesses richer and more delicate band structure tunability[25]. Under zero displacement field, the band structure of TLG can be decomposed to a combination of MLG and BLG (with hopping parameters that are different from the actual monolayer and bilayer systems), this fact by no means implies that the physics of TLG is a trivial repetition of MLG and BLG. In the presence of a magnetic field, the LLs originate from the MLG and BLG parts are not separated in energy but intersect with each other. If a vertical displacement field is introduced, the decomposition is no longer valid as the MLG and BLG parts hybridize. Each single-particle eigenstate is a superposition of the solutions in the non-relativistic

(NR) Landau problem and its weights in different NR levels vary with external fields. The separations between LLs can be tuned to generate many different orderings. For the 5/2 state in GaAs[4], the NR second LL is sandwiched between the lowest and third LLs. In contrast, one level in TLG that is similar to the NR second LL may be surrounded from above and below by other levels that have various different orbital contents. The LL mixing between them is very sophisticated and may lead to intricate competition between strongly correlated states.

## Results

The structure of our TLG devices is depicted in Fig. 1a, where two graphite gates are separated from TLG by insulating hBN layers (see Supplementary Fig. 1 for optical images of our device). By applying voltages $V_{tg}$ on the top gate and $V_{bg}$ on the bottom gate, the carrier density $n$ and the displacement field $D$ can be tuned independently as: $n = (C_b V_{bg} + C_t V_{tg})/e$ and $D = (C_b V_{bg} - C_t V_{tg})/2$, where $C_b$, $C_t$ are average geometric capacitances for the bottom and top gates. The lattice structure of TLG is shown in Fig. 1b together with the Slonczewski-Weiss-McClure (SWMc) parameters in its tight-binding description[26]. The potential difference between the top and bottom layers caused by the $D$-field is denoted as $2\Delta_1$. An additional variable $\Delta_2$ was proposed to characterize the intrinsic charge imbalance between the outer and middle layers[25]. The low-energy band of TLG for $D = 0$ mV/nm is shown in Fig. 1c, where MLG-like linear and BLG-like quadratic components can be discerned. When a perpendicular magnetic field $B$ is applied, the

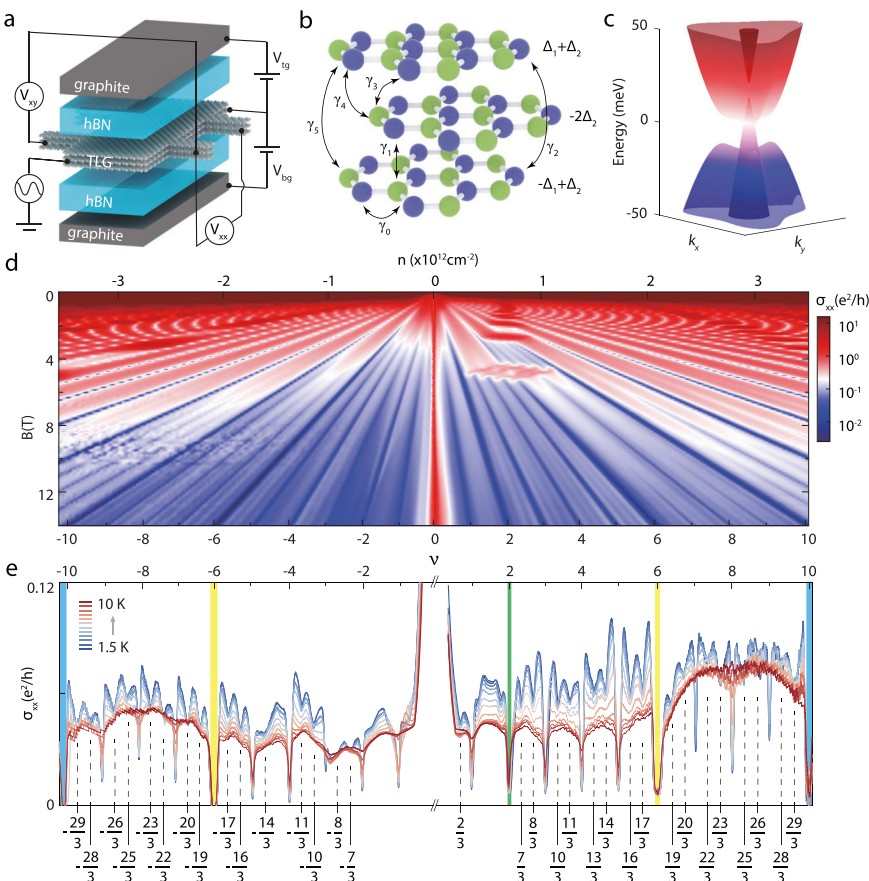

**Fig. 1 | Band structure of TLG and its quantum Hall states. a** Schematic of the TLG device with graphite top-gate and bottom-gate isolated by hBN. **b** The crystal structure of Bernal stacked TLG and the SWMc hopping parameters in its tight-binding model, here $\Delta_1$ and $\Delta_2$ describes potential difference between layers induced by the applied displacement field or non-uniform charge distribution respectively. **c** The low-energy band structure of TLG without the displacement field $D$ in the vicinity of the $\mathbf{K}_+$ valley. **d** The color map of the longitudinal

conductivity $\sigma_{xx}$ plotted versus carrier density $n$ and magnetic field $B$ at 1.5 K and $D = 0$ mV/nm. The filling factors defined at $B = 14$ T are given below the bottom axis. The diamond pattern at $B \approx 5$ T is attributed to level crossings. A plethora of FQH states are observed above 10 T. **e** $\sigma_{xx}$ as a function of $\nu$ for different temperatures at $B = 14$ T and $D = 0$ mV/nm. The filling factors $\nu = 2, \pm 6$ and $\pm 10$ are marked by green, yellow, and blue shaded regions, respectively. In the range $-6 \leqslant \nu \leqslant 6$, there are two MLG levels and four BLG levels for each spin projection (see Fig. 3f).

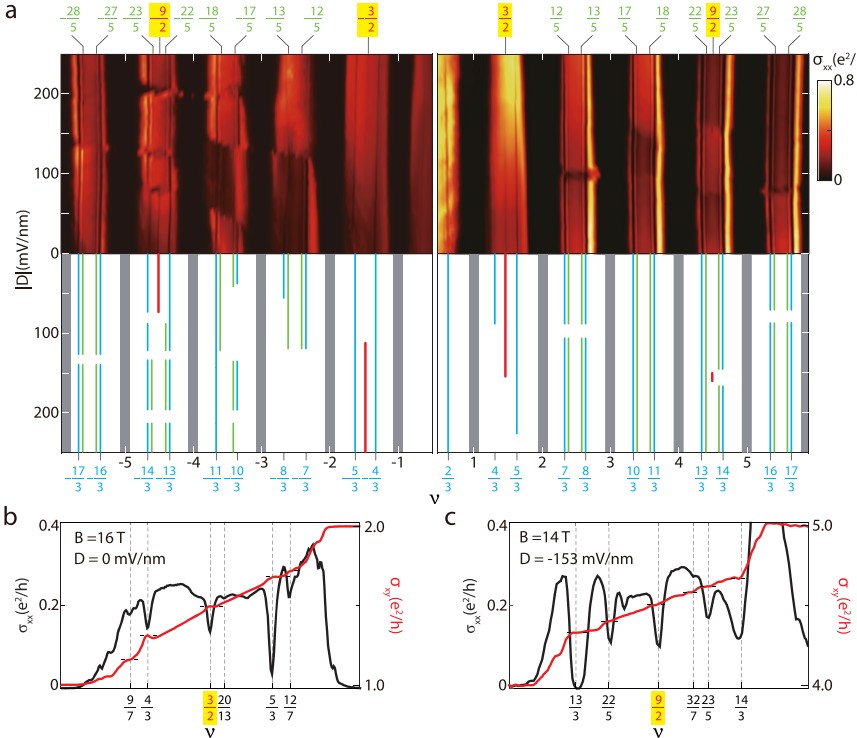

**Fig. 2 | Even- and odd-denominator FQH states in the filling factor range −6 ≤ ν ≤ 6. a** Top panel: the color map of the longitudinal conductivity $\sigma_{xx}$ plotted versus ν and the displacement field *D*. The magnetic field is *B* = 14 T and the base temperature is 15 mK. FQH states with both even and odd denominators are observed at various filling factors ν = *p/q* (*p* and *q* are integers) as signified by minima of $\sigma_{xx}$. The FQH states exhibit intricate evolution with *D*. Bottom panel: sketch map of the top panel in which the FQH states are marked for clarity. Grey shaded regions label the integer quantum Hall states. Red, blue, green lines label the FQH states with *q* = 2, 3, and 5, respectively. The line plots of $\sigma_{xx}$ and the Hall conductivity $\sigma_{xy}$ for 1 ≤ ν ≤ 2 with *B* = 16 T and *D* = 0 mV/nm (**b**) and 4 ≤ ν ≤ 5 with *B* = 14 T, *D* = −153 mV/nm (**c**). A few FQH states are indicated by the dashed lines and the associated values of ν are given below. The short horizontal lines mark the quantized plateau of $\sigma_{xy}$.

linear and quadratic bands give rise to two sets of LLs that scale as $\sqrt{B}$ and *B*, respectively. Some levels from these two sets may intersect with each other when the *B*-field varies. This can be seen in Fig. 1d where the Landau fan diagram measured at 1.5 K is presented. A few level crossings are observed around *B* = 5 T for 2 ≤ ν ≤ 6 as reported in previous works[24,27,28]. When the *B*-field increases to 11 T, longitudinal conductivity $\sigma_{xx}$ minima are observed at all integer filling factors −10 ≤ ν ≤ 10, which signifies a complete lifting of the spin and valley degeneracies of the LLs. Meanwhile, due to the high quality of the devices, a variety of well-developed FQH states emerge. The temperature dependence of the $\sigma_{xx}$ curves at 14 T are displayed in Fig. 1e for many FQH states with denominator 3, whose thermal activation behaviour clearly demonstrates their incompressible nature. An interesting feature is that some states in the range 6 ≤ ν ≤ 10 have different onset magnetic fields. FQH states at ν = 19/3, 23/3, 25/3, 29/3 are observed even before the neighbouring integer states at ν = 7, 9 appear (see Supplementary Fig. 2 for $\sigma_{xx}$ in the ν − *B* plane). In contrast, FQH states begin to develop at ν = 20/3, 22/3, 26/3, 28/3 near 10 T, at which point LL degeneracies at ν = 7, 9 have already been lifted. This phenomenon suggests that some FQH states are intimately connected with the unusual lifting of spin-valley degeneracy at ν = 7, 9.

Next we investigate the FQH states with −6 ≤ ν ≤ 6 at lower temperatures in detail. The color map of $\sigma_{xx}$ versus ν and *D* is plotted in Fig. 2a with the left (right) panel of the top row showing the hole (electron) side. The FQH states are summarized in the bottom row for clarity (see Supplementary Fig. 3 for the complete map). For various filling factors of the form $\tilde{\nu} = \nu − [\nu] = s/(2s + 1)$ and $1 − s/(2s + 1)$ ([ν] is the greatest integer less than or equal to ν and *s* = 1, 2), FQH states give rise to the observed minima of $\sigma_{xx}$. These states are illustrated as blue and green lines in the bottom sketch panel. Besides these odd-

denominator states, even-denominator FQH states are unambiguously seen at ν = −9/2, −3/2, 3/2, 9/2 and highlighted by red lines. The ν = −9/2 and 3/2 states can be realized at zero *D*-field and remain stable over a wide range of *D*, while the ν = −3/2 and 9/2 states only appear when a finite *D*-field is applied. In fact, the ν = 9/2 state can only be observed in a very narrow range of *D*. We plot $\sigma_{xx}$ together with the Hall conductivity $\sigma_{xy}$ around ν = 3/2 in Fig. 2b and the same quantities around ν = 9/2 in Fig. 2c. The concomitant appearance of exponentially suppressed $\sigma_{xx}$ and quantized $\sigma_{xy}$ unambiguously demonstrate that FQH states are realized at many different fractions. Interestingly, a weak $\sigma_{xx}$ minimum can be seen at ν = 20/13 close to the half-filling state at ν = 3/2, which could be a composite fermion state or a Pfaffian daughter state[29]. As shown in Supplementary Fig. 4, similar features are also observed in the vicinity of ν = −9/2, −3/2. In contrast, there is no signature of daughter states associated with ν = 9/2, and FQH states are observed clearly at $\tilde{\nu}$ = 2/5 and 3/5.

For a fixed displacement field, the dependences of $\sigma_{xx}$ on *B* around ν = −9/2, −3/2, 3/2 are presented in Fig. 3a–c. The existence of odd-denominator four-flux FQH states (ν = −12/7, −9/7, 12/7) underscores the high quality of our sample. The three even-denominator FQH states are robust in a considerable range of magnetic field. The line plot of $\sigma_{xx}$ at *B* = 16 T with −5 ≤ ν ≤ −4 is shown in Fig. 3d on which a few FQH states are indicated. The energy gaps of some states in Fig. 3d are deduced from their thermal activated behaviour and presented in Fig. 3e (see Supplementary Fig. 5 for the fitting). For the odd-denominator ones at $\tilde{\nu}$ = 2/5, 3/7, 4/9, the data can be understood using the composite fermion theory. A remarkable prediction of this theory is that the energy gap decreases to zero linearly as the filling factor $\tilde{\nu}$ approaches 1/2[30]. The gap values are fitted using $\Delta = \hbar e B_{\text{eff}}/m_{\text{CF}}$ in Fig. 3e, where $B_{\text{eff}} = (1 − 2\tilde{\nu})B$ is the effective magnetic field for

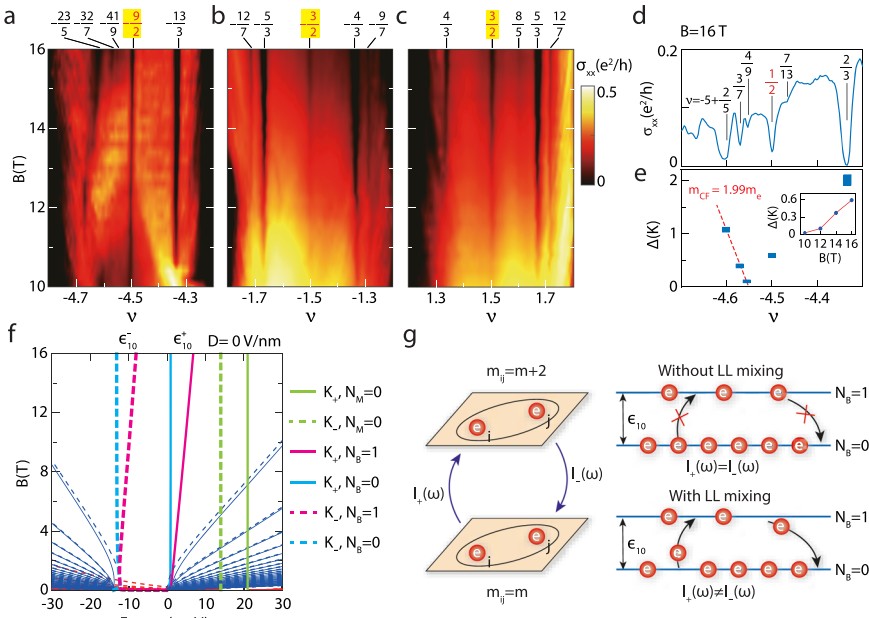

**Fig. 3 | Evolution of FQH states with magnetic field. a–c** The color maps of the longitudinal conductivity $\sigma_{xx}$ plotted versus $B$ and $\nu$ at 15 mK. The displacement field $D$ is zero in **a** and **c** and 217 mV/nm in **b**. **d** $\sigma_{xx}$ as a function of $\nu$ at $B = 16$ T. The vertical lines and the numbers mark some FQH states. **e** The energy gaps of the FQH states marked in **d** deduced from thermal activation measurements. The data points are drawn as blue rectangles with heights proportional to the uncertainty in Arrhenius fitting. The gaps for three odd-denominator states are fitted linearly using the red dash line $\Delta = \hbar e B_{\mathrm{eff}}/m_{\mathrm{CF}}$ ($B_{\mathrm{eff}} = (1 - 2\tilde{\nu})B$ is the effective magnetic field for composite fermions, and $m_{\mathrm{CF}}$ is composite fermion mass). Its inset shows the gap at $\nu = -9/2$ for different $B$. **f** The LLs computed theoretically at $\Delta_1 = \Delta_2 = 0$. The

$K_+$ ($K_-$) valley is represented using solid (dashed) lines and the spin degree of freedom is neglected for simplicity. The scheme of six levels is shown on the right of this panel. The MLG and BLG levels are marked by the subscripts M/B and their orbital contents are given by the numbers 0/1. The splitting between the two BLG levels in the same valley is denoted as $\epsilon_{10}^{\pm}$. **g** Schematic of the chiral graviton spectral functions $I_{\pm}(\omega)$. Left panel: Chiral gravitons are excited when the relative angular momentum of electron pairs $m$ is changed by 2. $I_+(\omega)$ [$I_-(\omega)$] is the spectral function of the operator that increases (decreases) $m$. Right panel: Particle-hole symmetry within one LL ensures that $I_+(\omega) = I_-(\omega)$, but LL mixing causes asymmetry between them. $\epsilon_{10}$ denotes the gap between the two LLs under consideration.

composite fermions and $m_{\mathrm{CF}} = 1.99\, m_e$ is the composite fermion mass. It is obvious that this rule is violated by the $\nu = -9/2$ FQH state whose energy gap is well above zero. The inset of Fig. 3e shows the evolution of the gap at $\nu = -9/2$ with $B$. It gets larger when $B$ increases as one would expect for an interaction driven state.

To understand the even-denominator states, we first inspect the LLs of TLG presented in Fig. 3f. The spin degree of freedom is neglected for simplicity. One simply assumes that all these levels are for spin-up electrons and bear in mind that each level has a spin-down counterpart. The parameter $\Delta_2$ is fixed at zero in most parts of our discussion, but a small value would not be detrimental either. In general, the single-particle eigenstates are six-dimensional vectors consist of NR Landau orbitals. For one eigenstate of the TLG LL, the largest weight may reside in the NR lowest LL, then it would be denoted as NR0. If there is a substantial weight in the NR second LL, it would be denoted as NR1. If there is no displacement field, the LLs can be divided to MLG and BLG ones. For $B \gtrsim 8$ T and $-6 \leqslant \nu \leqslant 6$, there are three levels in each of the $K_{\pm}$ valley and are further distinguished by the symbol $N_{M/B} = 0,1$. The subscript M/B traces the MLG/BLG origin of a level and the number 0/1 indicates that it is of the NR0/NR1 type. It is well-known that the NR second LL is favorable for realizing FQH state at half filling, as exemplified by the 5/2 state in GaAs[4]. Using this information, we can provide a simple picture for the $\nu = -9/2$ and 3/2 states without the $D$-field. As shown in Fig. 3f, electrons fill all the LLs below $K_-, N_B = 0$ at $\nu = -6$. An extra 3/2 filling of electrons are added to arrive at $\nu = -9/2$. It is natural that the spin-up $K_-, N_B = 0$ level is fully occupied first. The remaining 1/2 may enter the spin-down $K_-, N_B = 0$ level or the spin-up $K_-, N_B = 1$ level. Both single-particle and interaction effects should be incorporated to determine which scenario is realized. The Zeeman splitting is $E_Z = 1.62$ meV at 14 T whereas the separation $\epsilon_{10}^-$ between

$K_-, N_B = 1$ and $K_-, N_B = 0$ is 4.59 meV. If there is no interaction, electrons would populate the spin-down $K_-, N_B = 0$ level. However, it is has been found that maximal spin polarization is favoured by Coulomb interaction in many cases, so the electrons may instead occupy the spin-up $K_-, N_B = 0,1$ levels in a certain parameter regime. This leads to a half-filled NR1 level in which the Moore–Read type states could emerge. A similar picture has also been proposed to understand some even-denominator states in BLG[17,18].

This picture for $\nu = -9/2$ is corroborated using exact diagonalization results on the torus. There are six quasi-degenerate ground states (Supplementary Fig. 6), which is consistent with the prediction for the Moore–Read type states as well as previous results in BLG[31–34]. This result alone cannot tell us if the state is of the Pfaffian, anti-Pfaffian, or particle-hole symmetric Pfaffian type. To this end, we have computed the chiral graviton spectral functions[35,36]. As illustrated in Fig. 3g, these quantities are designed to reveal the relative angular momentum of electron pairs. It has been shown that the dominant chirality is negative (positive) for the Pfaffian (anti-Pfaffian) wavefunction[36]. If we only keep the $K_-, N_B = 1$ level in our calculation, particle-hole symmetry ensures that the two chiralities are the same. After incorporating LL mixing with the $K_-, N_B = 0$ level (and excluding all other levels), the Pfaffian state becomes the favored one (Supplementary Fig. 7). This is consistent with the possible existence of a daughter state at $\nu = -5 + 7/13$. In general, when a NR0/NR1 doublet has 3/2 filling, one may expect to see an even-denominator FQH state. For the $\nu = 3/2$ state, the levels that should be considered are $K_+, N_B = 0$ and $K_+, N_B = 1$. It should also be of the Pfaffian type in view of the weak minima at $\nu = 20/13 = 1 + 7/13$. For the range of $B$ that have been studied, it may be sufficient to keep these two levels, but we should bear in mind that the MLG levels are not too far away in energy. If $B$ increases

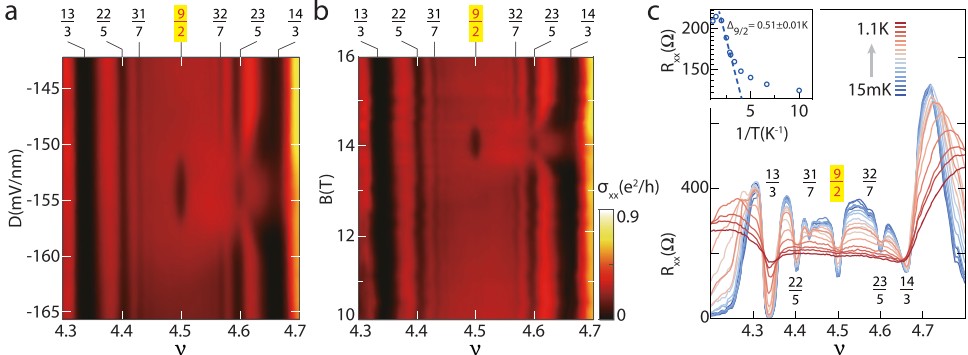

**Fig. 4 | Displacement and magnetic field-tuning of FQH states in the filling factor range $4 \leqslant \nu \leqslant 5$.** The color map of longitudinal conductivity $\sigma_{xx}$ plotted versus $D$ (with $B = 14$ T fixed in **a**) or $B$ (with $D = -153$ mV/nm fixed in **b**) and $\nu$ at 15 mK. Phase transitions of FQH states are observed at $\nu = 9/2, 32/7, 23/5$. **c** The temperature dependence of $\sigma_{xx}$ versus $\nu$ for $D = -153$ mV/nm and $B = 14$ T. Its inset shows the Arrhenius fitting of the thermal activation gap at $\nu = 9/2$.

to about 29 T, the $\mathbf{K}_+, N_B = 1$ level would be equidistant from $\mathbf{K}_+, N_B = 0$ and $\mathbf{K}_+, N_M = 0$. This results in exotic LL mixing that has not been explored in previous works and the fate of the 3/2 state would be very interesting. A higher magnetic field is required to do the same thing for the $\mathbf{K}_-$ valley.

When the displacement field is turned on, the FQH states evolve in different manners. Rigorously speaking, the Landau levels can no longer be labeled using $N_{B/M}$ etc. However, it is convenient to track the evolution of each level with $D$ and refer to them using the names at $D = 0$ mV/nm. The disappearance of the $\nu = -9/2$ and 3/2 states can be explained qualitatively based on Supplementary Fig. 8b. The $\mathbf{K}_-, N_B = 0$ level goes down when $D$ increases and eventually crosses with two NR2 type levels. In this regime, the $\nu = -9/2$ state would not correspond to a half-filled NR1 type level, so no even-denominator FQH state is expected. The experimental data in Fig. 2a indeed suggests that a level crossing occurs at $D \approx 80$ mV/nm. For the $\nu = 3/2$ state, the $\mathbf{K}_+, N_B = 0$ and $\mathbf{K}_+, N_B = 1$ levels cross with each other at sufficiently large $D$ and the weight of the NR1 orbitals in $\mathbf{K}_+, N_B = 1$ gradually decreases. It is thus plausible that the gradually weakens as $\epsilon_{10}^+$ decreases, which is consistent with the experimental data in Fig. 2a. In contrast to these two filling factors, a state emerges at $\nu = -3/2$ when $D$ becomes sufficiently large. In the absence of the $D$-field, the spin-up $\mathbf{K}_+, N_B = 0$ level is half filled and no FQH state is expected. The $\mathbf{K}_-, N_B = 1$ level moves up with $D$ and crosses with the $\mathbf{K}_+, N_B = 0$ and $\mathbf{K}_+, N_B = 1$ levels at quite large $D$. This could occur at smaller $D$ if a small positive $\Delta_2$ is invoked (see Supplementary Fig. 8c). After the crossing, the electrons fully occupy the spin-up and spin-down $\mathbf{K}_\pm, N_B = 0$ levels at $\nu = -2$. The $-3/2$ state would correspond to a half filled NR1 level as indicated in Supplementary Fig. 8d. It is likely of the Pfaffian type given the weak feature observed at $\nu = -2 + 7/13$ (see Supplementary Fig. 4).

Finally, we study the $\nu = 9/2$ state that only appears in a very small window of $D$ and $B$. The color map of $\sigma_{xx}$ around $\nu = 9/2$ is presented in Fig. 4a, b (with one parameter fixed and the other varied). As the $\nu = 9/2$ state emerges, the gap at $\nu = 23/5$ closes and then reopens. This implies that a phase transition has occurred between two different $\nu = 23/5$ states. The $\nu = 32/7$ state simply disappears when the $\nu = 9/2$ state is observed. On the contrary, the states at $\nu = 22/5$ and $31/7$ remain stable and no transition is found. By fitting the thermal activation data in Fig. 4c, the gap at $\nu = 9/2$ is found to be ~ 0.51 K. For our tight-binding model with $\Delta_2 = 0$, the active levels at $\nu = 9/2$ are of the NR0 type, which is unfavorable for realizing even-denominator states. If we change $\Delta_2$ to $-10$ meV, the Pfaffian state could be realized at $\nu = 9/2$ (see Supplementary Fig. 9 and 10). This analysis is not very satisfactory because such a value of $\Delta_2$ is not quite reasonable and it is difficult to explain why this state only appear in a narrow range of $D$. To this end, we may consider multi-component FQH states whose spin and/or valley indices are not polarized. The Halperin 331 state is a well-known

two-component state at $\nu = 1/2$[37]. Another example is the Jain state constructed from the parton theory[38,39]. For a suitable range of $D$, two NR0 type levels are almost degenerate. At $\nu = 9/2$, two NR0 type levels with the same valley index are half filled. The interaction between different valleys may be altered by valley anisotropic terms[40] to stabilize the Jain state[41]. This mechanism was proposed to explain the 1/2 state observed in MLG[19], and a more detailed analysis is needed to check if it also works for TLG.

## Discussion

In summary, our results reveal the rich odd- and even-denominator FQH states in TLG and underscore its extraordinary tunability due to intricate interplay of spin, valley, and orbital degrees of freedom. While the odd-denominator states are most acceptably described by the composite fermion theory, other candidates such as the Read-Rezayi states with non-Abelian Fibonacci anyons[42] are also possible and deserve further investigations. On the other hand, it would be fruitful to further explore the consequences brought by the evolution of LL eigenstates and their mixing and crossings. By varying external fields, we may switch between multiple Abelian and non-Abelian FQH states as well as non-FQH states, and continuous quantum phase transitions may be feasible in some of these process. Since FQH states are not described by the Landau paradigm based on symmetry breaking, their transitions are quite likely not captured by the standard Landau–Ginzburg–Wilson theory[43]. The low-energy effective theory of FQH states generally involve Chern–Simons gauge fields[44], so there could be many exotic transitions described by strongly coupled quantum field theory.

## Methods

The devices were fabricated using our "pick-up method"[45] to achieve a multi-layer heterostructure with the TLG encapsulated by two flakes of hexagonal boron nitride (hBN) and thin graphite flakes as the top and bottom gates. The stacks were annealed under a high vacuum at 350 °C for 2 h. Electron-beam lithography was used to write an etch mask to define the Hall-bar geometry and the electrodes. Redundant regions were etched away by $CHF_3/O_2$ plasma[45]. Finally the TLG and gates were edge-contacted[45] by e-beam evaporating thin metal layers consisting of Cr/Pd/Au (1 nm/15 nm/100 nm).

The transport measurements were performed in two systems, a dilution fridge with a base temperature of 15 mK and a VTI fridge down to 1.5 K, and both are with superconducting magnets. All data were taken using the standard four-terminal configuration with lock-in amplifier techniques by sourcing an AC current $I$ between 10 and 100 nA at a frequency of 17.777 Hz. The data of longitudinal conductivity $\sigma_{xx}$ and Hall conductivity $\sigma_{xy}$ are obtained from the measured resistances by $\sigma_{xx} = \rho_{xx}/(\rho_{xx}^2 + R_{xy}^2)$ and $\sigma_{xy} = R_{xy}/(\rho_{xx}^2 + R_{xy}^2)$.

## Data availability

The data that support the findings of this study are available from the corresponding authors upon request.

## Code availability

The code that support the findings of this study is available from the corresponding authors upon request.

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

## Acknowledgements

L.W. acknowledges support from the National Key Projects for Research and Development of China (Grant Nos. 2021YFA1400400, 2022YFA1204700), National Natural Science Foundation of China (Grant No. 12074173) and Natural Science Foundation of Jiangsu Province (Grant No. BK20220066). Y.H.W. acknowledges support from the National Natural Science Foundation of China (Grant No. 12174130). Z.Z. acknowledges the National Natural Science Foundation of China (Grant No.12074375), the Strategic Priority Research Program of CAS (Grant

No.XDB33000000). K.W. and T.T. acknowledge support from the JSPS KAKENHI (Grant Numbers 21H05233 and 23H02052) and World Premier International Research Center Initiative (WPI), MEXT, Japan. F.C.Z. acknowledges partial support from China Ministry of Sci and Tech (grant 2022YFA1403902), Priority Program of CAS grant No XDB28000000, NSFC grant No 11674278, and Chinese Academy of Sciences under contract No. JZHKYPT-2021-08. G.L. acknowledges the support by the NSFC of China (grant No. 9206520), the National Basic Research Program of China from the MOST (grant No. 2022YFA1602803), and the Strategic Priority Research Program of the Chinese Academy of Sciences (grant No. XDB33010300). This work is supported by the Synergic Extreme Condition User Facility and by the Innovation Program for Quantum Science and Technology (Grant No. 2021ZD03026001).

## Author contributions

L.W. conceived and designed the experiment. Y.C., Y.H., and Q.L. fabricated the samples. Y.C., Y.H., Q.L., G.K., C.X., B.T., G.L., and L.L. performed the transport measurements. Y.C., L.W., Y.H.W., F.C.Z., and Z.Z analyzed the data. Y.H.W. conducted theoretical analysis. K.W. and T.T. supplied the hBN crystals. Y.C., Y.H.W. and L.W. wrote the manuscript with input from all co-authors.

## Competing interests

The authors declare no competing interest.
