## [Peer Review File · Nature Communications]

REVIEWER COMMENTS

Reviewer #1 (Remarks to the Author):

The authors present an observation of even and odd denominator fractional quantum Hall states in a dual-gated ABA trilayer graphene system. The high quality of the sample enables the full splitting of Landau levels across a wide range of filling factors, from -10 to 10, revealing multiple fractional QH states. Investigating the dependency of these states on both magnetic and displacement fields, they discover that certain fractional states can be manipulated, switched on or off, through the displacement field. Qualitatively explaining this tunability, the authors discuss the LL crossing within the single-particle picture of ABA trilayer graphene, comprising MLG-like and BLG-like subbands.

While prior studies have examined even and odd denominator fQH in monolayer (MLG) and bilayer (BLG) graphene, this report marks the first demonstration of fQH in ABA trilayer graphene. The manuscript not only presents a rich phase diagram but also offers clear experimental data, likely sparking significant interest within the graphene research community. Therefore, I recommend the publication of this manuscript in Nature Communications.

I have a few remarks that the authors should address.

1) In extended data fig. 6a, why is $\nu = -9/2$ positioned at NR1 (dashed pink line) rather than at NR0 (dashed blue line)? It seems there might be a misalignment.

2) The $\nu = -9/2$ and $\nu = -3/2$ states appear to stem from K^- , $NB=0$, and K^+ , $NB=0$ states within the BLG subband, respectively. However, in reference 18 for bilayer graphene, even denominator states arise from $NB=1$ states. Considering these states originate from the same BLG band, should they not emerge from the same NB ?

3) The last paragraph of page 6 becomes somewhat challenging to follow when the authors elucidate the effect of the displacement field using Extended Data Fig. 6b. In the main text, the authors utilize K^- , NB or K^+ , NB notation, whereas Extended Data Fig. 6b employs NR0 or NR1. Consistency in notation between the main text and Extended Data Fig. 6b would enhance clarity.

4) The authors mention that for the $\nu = 3/2$ state, the K^+ , $NB=0$ and K^+ , $NB=1$ levels are partially occupied. However, according to Fig. 3f, shouldn't it be that K^+ , $NB=0$ is fully filled and K^+ , $NB=1$ is partially filled? Clarification on this point would be appreciated.

Reviewer #2 (Remarks to the Author):

This manuscript reports the observation of even-denominator (ED) fractional quantum Hall (FQH) states in ABA ‘bernal’ trilayer graphene. Even-denominator states have been of great interest for some time due to the possibility of using certain such states to observe and control nonabelian anyons. However, not all even-denominator ground states support such phenomena. In monolayer graphene, even-denominator states that are not expected to support nonabelian anyons arise due to the roles of the spin and valley degrees of freedom and are thus referred to as ‘multicomponent’. On the other hand, in bilayer graphene, even-denominator states that so far appear to be consistent with those predicted to exhibit nonabelian anyons have been observed. These states may be referred to as ‘single-component’ in that they are predicted to be spin and valley polarized, and the origin of the even denominator is thus ‘pairing’ between Landau levels (LLs) with different indices. Given this situation, ABA trilayer graphene is an interesting system for exploring this kind of physics because its band structure consists of coexisting massless monolayer-like and massive bilayer-like bands whose hybridization can be tuned via the electric field. The authors observe four even-denominator states, of which they attribute three to the bilayer-like massive bands and one to the monolayer-like massless bands. This raises the possibility that the three states observed in the massive bands may support nonabelian anyons.

From my point of view this manuscript is suitable for publication in Nature Communications in its present form. In terms of the technical side of the experiment in my mind there is no question that everything is of high quality and that the identification of the states and related data interpretation is reliable. The observations are new, and this data set on QHE in ABA trilayer is to my knowledge the highest quality available in the literature. What is especially interesting in this work is that this is the first platform that may have the possibility of ‘tuning’ between potentially nonabelian versus multicomponent ED states in a single device. This feature would seem to open lots of doors for future work, in particular probing the isospin order in the various ground states or studying more complex geometries. For example one could imagine making devices with layer discontinuities (e.g. at a trilayer/bilayer interface) which would presumably be useful for probing the properties of the incompressible state. Because of the new possibilities it raises I think the work is novel and notable enough to warrant publication.

There is one aspect I would like to ask about and that the authors could perhaps expand on in the manuscript if they think it is warranted. How well established is the level ordering depicted in Fig. 3f? It is of course difficult to directly determine the spin and especially orbital structure of various ground states, but in particular, is it really established that the claimed orbital splitting indeed exceeds the Zeeman splitting at high fields (~ 4.6 meV claimed by the authors versus ~ 1.6 meV at 14 T)? Even indirect experimental evidence would be helpful as one would like to be sure what role the spin is playing.

Reviewer #3 (Remarks to the Author):

Report

The paper presents experimental observations of FQHE in trilayer graphene versus magnetic field strength and vertical voltage (called as a displacement field). Though the bilayer graphene has been explored relatively widely with regard to quantum Hall behavior, the trilayer one was not a subject for such measurements in the past. Thus the experimental study of trilayer graphene is of large significance.

The results are interesting but the interpretation of experimental data and theoretical comments must be revised, as are frequently confusing. The quality of colorful figures is guaranteed by plotters of experimental facilities, but the commentary part developed by Authors is insufficient and partly wrong if compared to former studies of bilayer graphene, especially in higher LLs.

To be more specific the following phrases/issues must be revised:

“And fully filling such Landau levels (LLs) one by one gives rise to the integer quantum Hall states [1]. Within one LL, the strong Coulomb interaction dominates over kinetic energy in the highly degenerated LL flatband, and further conduces to the emergence of FQH states at certain fractional fillings ν [2].”

Both these phrases are incorrect. In fact both IQHE and FQHE arise due to strong correlations of interacting 2D electrons at the magnetic field presence. The correlations are induced by the interparticle interaction in 2D geometry. Important is a cyclotron commensurability of braid trajectories with Wigner lattice of repulsing electrons deposited on positive planar jellium. The commensurability conditions are topological invariants robust against local imperfections and small temperature chaos (thus IQHE and FQHE are observable at low temperatures, even though the

perfect classical Wigner lattice is defined for $T=0$ K). The patterns of the cyclotron commensurability (topological invariants) are discrete versus magnetic field value and define a general hierarchy of filling fractions embracing fillings ratios at which IQHE and FQHE manifest

themselves. Correlations lead to an energy gain of multiparticle collective system of all electrons (counted with respect to noninteracting multiparticle system) and IQHE (similarly as FQHE) does not exist for non-interacting electrons (as Wigner lattice requires electron repulsion, though the interaction does not enter here in a perturbative manner). The complete fillings of LLs are possible also for gaseous systems of fermions, which do not exhibit, however, IQHE (the coincidence of the correlated wave function at

$\nu=1$ with Vandermonde polynomial for non-interacting fermions is a source of frequent confusion).

Thus fully filling of LLs does not give rise to the IQHE (despite a common but false popular opinion – IQHE is not a single-particle phenomenon, but is a collective topological effect of interacting electrons, similar as FQHE) – cf. [b] *Annals of Physics* 430 (2021) 168493.

Next, It must be emphasized that the old model of composite fermions (CFs) is obsolete and incorrect in view of many new experiments. CF model fails in explanation of FQHE in so-called enigmatic states in the LLL (like $5/13$, $4/11$, $3/10$, $3/8$ and many others), in higher LLs of GaAs – cf. [a] (FQHE states at fractions with denominator 3, as those reported in the paper, are not of CF type in higher LLs), in bilayer graphene – cf. [d], in two layer GaAs systems both with [i,j,a] and without [a] carrier

interlayer hopping, in monolayer graphene at $\nu=1/2$, $1/4$ [g], at $1/2$ state in bi-layer graphene in suspended sample [e], in hole 2DHS GaAs at $\nu=3/4$ [k] and in many others. Such an abundance of

counterexamples against CF model arises from a wrong idea about effective quasiparticle with artificial fluxes attached. None fluxes are attached in fact to electrons (a hypothetical magnetic field, quanta of which are assumed to be pinned to CFs, does not exist – this field is only an auxiliary fictitious construction to mimic additional loops in 2D multiloop braids, and in addition CF model does it not correctly). It has been demonstrated [b] that CF model confuses an important parameter in the Jain hierarchy $\nu=n/((q-1)n + (-1)^n)$, n positive integer – the ratio of a total number of electrons in 2D system to a number of next nearest neighbors of consecutive rank in Wigner lattice with the

Landau index in artificial and auxiliary spinless gaseous fermion system. Though the latter is also an integer, not all integers are in fact possible, due to the structure of next-next-nearest neighbors in Wigner lattice, which was not taken into account in CF model. The wave functions in CF model assumed as projected from higher LLs (in the spinless auxiliary LL model) onto LLL (to remove singularities) are wrong compared to the true multiparticle wave functions obligatory keeping the braid symmetry [b]. Therefore referring to CFs in the paper is confusing. Next, the Ref. [20] cited in the paper presents a naive attempt to lift the CF model to account bilayer Hall systems (in bilayers we deal with the natural distribution of loops of multiloop cyclotron braid orbits among layers,

which authors of [20] try to model by strange division of flux quanta attached to electrons).

Similarly, in a series of recent CF continuation papers promoted in APS journals and trying to introduce next artificial constructions to CFs like CF interaction or “partons” (probably to mimic loops in multiloop

braids) is a rather incorrect interpretation favoured by lobby of supporters, which still in a biased manner protects CF model despite its evident failure in view of majority of Hall experiments and internal inconsistency in the CF model.

Authors address also to $5/2$ state (which actually is of paired type also in topological approach, confirmed by exact diagonalization in toy models, on the other side) and to so-called “non-Abelian anyons”. As reported in the paper the state $9/2$ may also be of paired character (but $1/2$ and $3/2$ not, what is clear in the braid approach), such a comment needs, however, a more clear explanation. “Non-Abelian anyons” are not quasiparticles, they are different notion than Abelian anyons. The latter are particles satisfying a fractional statistics – non-Abelian anyons are linked to matrix $n \times n$ ($n > 1$) unitary representations of the braid group and are related to a possible degeneracy of excitations transforming along such matrices. They are not “protected by topological invariants”, but may serve to approximate (if are sufficiently dense) any unitary transformation of an universal gate for QIP according to Kitaev idea. Moreover, the idea of topological quantum computing cited by Authors bases on a model of an Abelian anyon on torus (but on torus scalar unitary representations of braid group – Abelian anyons – do not exist [1]). A false assumption makes all next implications logically true, which challenges to some extent the cited paper Ref. [15]. This must be clarified, how to understand non-Abelian anyons, if they are needed in the paper.

Finally Authors try to add a significance to their experimental observations addressing to the field theory (in summary) – this is also confusing a bit, as the field theory of Chern-Simons type is not fundamental – this is only a formal exemplification of artificial fluxes attached to fermions or bosons to change the statistics in 2D by hands on demand. Chern-Simons field is not a canonical transformation and despite the name of “gauge field”, the resulting shifted statistics (and auxiliary fluxes added to anyons or composite fermions) are not derived here from first rules but are by hands assumed/selected in the same artificial manner as for CF model. Over 30 years lasting story of such

an approach to FQHE without any significant progress evidences that CF (and related formulation in field theory terms) is not helpful, if the homotopy background of quantum Hall physics is neglected.

The paper has a chance to contribute to the domain of FQHE in unexplored as of yet region of trilayer graphene, but must take into account the current knowledge avoiding continuation of apparently misleading obsolete ideas (despite they are widespread and indiscriminately repeated in many papers mostly experimental, but are incorrect).

If Authors want to comment on the reason of observed FQHE features at filling ratios in higher LLs in trilayer structure, two facts must be included,

1) various patterns of the distribution of loops of multiloop braids among layers in extension of the behavior known from bilayer graphene [d] (in particular fractions with the denominator 5), 2) the size of cyclotron orbits in higher LLs larger than in the LLL [a], which is important for the

commensurability invariants. These two circumstances decide about the general hierarchy of homotopy patterns (filling fractions)

in multilayer 2D structures including higher LLs also in trilayer graphene. Note that not all patterns from the general hierarchy are observable, and the experimentally detected hierarchy can be tuned by various factors deciding on stability of particular patterns (stability is controlled by the envelope part of the multiparticle wave function dependent on single-particle LL wave functions contaminated by a specific crystal field in graphene and in multilayer structures – here is a place for various

symmetry breaking language).

The discussion of the results needs thus a major revision. In view of the above comments the formulation of the summary (and in many other places in the text) seems to be insufficient/missed in

part. Some elusive comments about various symmetry breaking as the source of complicated hierarchy structure do not have here a central significance – the complicated hierarchy of filling

ratios is a matter of homotopy invariants. The observed stability of particular homotopy patterns changing in response to magnetic field value (at the same filling ratio as in a fan diagram) or to

vertical electric field, does not concern the general hierarchy but only competition between various patterns available at same filling fractions (which is explained within topological homotopy approach [b], but not in CF model). This has been demonstrated earlier experimentally, like for the stability of FQHE at $1/2$ or $1/4$ in monolayer graphene at some window for magnetic field [g,f], stability of $1/2$ FQHE state in bilayer graphene [in suspended sample [e] though not in a sample deposited on hBN

substrate), stability of FQHE at $\nu=3/4$ in 2DHS in GaAs versus Hall metal at $\nu=3/4$ in 2DES in GaAs at similar other conditions [k], at the demonstrated transition of FQHE bilayer hierarchy to monolayer one by application of vertical electric field [h], cf. also SI to [d] for explanation. All these have been explained in homotopy terms, apparently beyond the CF model, which is unable to consider such situations.

The same here – in the paper Authors use the improper theoretical methodology (in fact only phraseology repeated from many former publications, where similar experiments were reported without, however, successful explanations if neglected the braid homotopy conditioning of FQHE hierarchy). In the case when the assumptions of CF model are fully clarified (including its limitations

[b]), CF model approach does not conform with the already gained knowledge and the paper must be revised to avoid confusions. The addressing to bizarre concepts of CF theory developments as an

interaction of CFs or partons is similar to producing epicycles to epicycles to fit reality of completely different character.

For example, some phrases on filling fraction denominators can be highly deceptive if addressed to higher LLs in multilayer structure (e.g., as shown in [a] the denominators 3 are not obligatory referred to three-loop orbit, unless in the LLL in GaAs; in higher LLs in GaAs [a] they are related to single-loop orbit precluding any CF – three loop – interpretation). In graphene the situation is even more complicated because of spin-valley subband structure and odd denominators [d] may not be related to a simple case in the LLL of GaAs, where CF intuitions might be accepted (though not for even denominator fractions or other enigmatic states).

Minor comments – lettering in Bibliography should be revised. In the sentence “A higher magnetic field is require to do the same thing for the K– valley.” “required” should be rather. In caption to Fig.1 “The filling factors defined at B = 14 T is given below the bottom axis.” It should be rather “are”. mentioned above references

a. J. Jacak and L. Jacak, The commensurability condition and fractional quantum Hall effect hierarchy in higher Landau levels. JETP Letters 102, 19 (2015).

b. J. Jacak, Topological approach to electron correlations at fractional quantum Hall effect, Annals of Physics 430, 168493 (2021).

c. J. Jacak, Superfluidity of indirect excitons vs quantum Hall correlation in double Hall systems: Different types of physical mechanisms of correlation organization in Hall bilayers, Phys. Lett. A 382, 41 (2018).

d. J. Jacak, Unconventional fractional quantum Hall effect in bilayer graphene, Scientific Reports, 7, 8720 (2017)

e. D. K. Ki, V. I. Falko, D. A. Abanin, A. Morpurgo, Observation of even denominator fractional quantum Hall effect in suspended bilayer graphene, Nano Lett. 14, 2135 (2014).

f. J. Jacak, Explanation of an unexpected occurrence of $\nu = \pm 1/2$ fractional quantum Hall effect states in monolayer graphene, J. Phys.: Condens. Matter 31, 475601 (2019).

g. A. A. Zibrov, E. M. Spanton, H. Zhou, C. Kometter, T. Taniguchi, K. Watanabe and A. F. Young,

Even denominator fractional quantum Hall states at an isospin transition in monolayer graphene, Nat. Phys. 14, 930 (2018)

h. P. Maher, L. Wang, Y. Gao, C. Forsythe, T. Taniguchi, K. Watanabe, D. Abinin, A. Papic, P. Cadden-Zimansky, J. Hone, P. Kim and C. R. Dean Tunable fractional quantum Hall phases in bilayer graphene. Science 345, 61 (2014)

i. Y. W. Suen, L. W. Engel, M. B. Santos, M. Shayegan, D. C. Tsui, Observation of a $\nu = 1/2$ fractional quantum Hall state in a double-layer electron system, Phys. Rev. Lett. 68, 1379 (1992)

j. P. Eisenstein, G. S. Boebinger, L. N. Pfeiffer, K. W. West, S. He, New fractional quantum Hall state in double-layer two-dimensional electron systems, Phys. Rev. Lett. 68, 1383 (1992)

k. C. Wang, A. Gupta, S. K. Singh, Y. J. Chung, L. N. Pfeiffer, K. W. West, K. W. Baldwin, R. Winkler, M. Shayegan. Even-denominator fractional quantum Hall state at filling factor $\nu = 3/4$. Phys. Rev. Lett., 129, 156801 (2022)

l. T. Einarsson, Fractional statistics on a torus, Phys. Rev. Lett. 64, 1995 (1990)

Reviewer #1 (Remarks to the Author):

The authors present an observation of even and odd denominator fractional quantum Hall states in a dual-gated ABA trilayer graphene system. The high quality of the sample enables the full splitting of Landau levels across a wide range of filling factors, from -10 to 10, revealing multiple fractional QH states. Investigating the dependency of these states on both magnetic and displacement fields, they discover that certain fractional states can be manipulated, switched on or off, through the displacement field. Qualitatively explaining this tunability, the authors discuss the LL crossing within the single-particle picture of ABA trilayer graphene, comprising MLG-like and BLG-like subbands.

While prior studies have examined even and odd denominator fQH in monolayer (MLG) and bilayer (BLG) graphene, this report marks the first demonstration of fQH in ABA trilayer graphene. The manuscript not only presents a rich phase diagram but also offers clear experimental data, likely sparking significant interest within the graphene research community. Therefore, I recommend the publication of this manuscript in Nature Communications. I have a few remarks that the authors should address.

Response: We thank the reviewer for this very positive feedback and for the enthusiastic recommendation of publication. Below we provide detailed answers to the specific questions.

1) In extended data fig. 6a, why is $\nu=-9/2$ positioned at NR1 (dashed pink line) rather than at NR0 (dashed blue line)? It seems there might be a misalignment.

Response: We thank the reviewer for this question. This confusion is caused by our careless presentation. Extended Data Fig. 6a is a theoretical calculation of the single-particle Landau levels without the spin degree of freedom. Because the Zeeman coupling can be incorporated trivially, we did not show it explicitly to avoid clutter in the figure. The integer quantum Hall state with filling factor -6 is simple. To obtain the -9/2 state, we should add 3/2 filling to the -6 state. It is most likely that the NR0 level with spin-down is fully occupied first. The 1/2 filling may correspond to two states: 1. the NR0 level with spin-up is half-filled and the NR1 level with spin-down is empty; 2. the NR0 level with spin-up is empty but the NR1 level with spin-down is half-filled. For non-interacting electrons, the former possibility is realized given the Zeeman energy and the splitting between NR0 and NR1 levels. However, electron correlation might select the latter one because maximal spin-polarization is energetically favorable. For example, it is well established that the 1/3 spin-polarized Laughlin state is realized in a spinful system even if the Zeeman energy is zero. As the reviewer has already noticed in the next question, a similar situation was found in bilayer graphene (BLG), as shown in Nature 549, 360 (2017) [see Fig. S6 and related discussions in the Supplemental Information]. We have made a few changes to address this question.

1) In the last paragraph of page 4, we wrote “The spin degree of freedom is neglected for simplicity. One simply assumes that all these levels are for spin-up electrons and bear in mind that each level has a spin-down counterpart.”

2) In the first paragraph of page 6, we wrote “It is natural that the spin-up $\mathbf{K}_{-,N_{\text{B}}=0}$ level is fully occupied first. The remaining $1/2$ may enter the spin-down $\mathbf{K}_{-,N_{\text{B}}=0}$ level or the spin-up $\mathbf{K}_{-,N_{\text{B}}=1}$ level. Both single-particle and interaction effects should be incorporated to determine which scenario is realized.”

3) In the captions of Fig. 3 in the main text and Extended Data Fig. 6, we added “the spin degree of freedom is neglected for simplicity.”

2) The $\nu = -9/2$ and $\nu = -3/2$ states appear to stem from K^- , $NB=0$, and K^+ , $NB=0$ states within the BLG subband, respectively. However, in reference 18 for bilayer graphene, even denominator states arise from $NB=1$ states. Considering these states originate from the same BLG band, should they not emerge from the same NB ?

Response: The reviewer made an insightful comment by making connections with bilayer graphene (BLG). As we have explained above, the $\nu=-9/2$ state most likely reside in the NR1 level (which is $N_B=1$ at $D=0$). This is consistent with the observations in BLG. The physics at $\nu=-3/2$ is more complicated. To understand it, it is helpful to begin with the $\nu=-2$ state. When Δ_1 and Δ_2 are both chosen to be zero, the $\nu=-2$ state is most likely obtained by filling the spin-up and spin-down NR0 and NR1 levels in the K^- valley (shown as dashed lines in Extended Data Fig. 6a). In this case, the $-3/2$ state corresponds to the half-filled NR0 level in the K^+ valley, so no FQH state is expected. For appropriate choices of Δ_1 and Δ_2 , the Landau levels from K^+ and K^- valleys cross (shown as solid and dashed lines in Extended Data Fig. 6c). At this point, the $\nu=-2$ state may instead be obtained by filling the spin-up and spin-down NR0 in both valley. In this case, the $-3/2$ state corresponds to the half-filled NR1 level in the K^+ valley, so a FQH state is expected. We have made a few changes to address this question.

1) In the first paragraph of page 6, we wrote “A similar picture has also been proposed to understand some even-denominator states in BLG [17,18].”

2) In Extended Data Fig. 6, we added a new panel (d) in which the $-3/2$ filling is indicated and the caption “Zoom-in view of the region enclosed by dashed black lines in panel **c**. Two possible locations of $\nu=-3/2$ are indicated.”

3) In the first paragraph of page 7, we wrote “The \mathbf{K}^- , $N_B=1$ level moves up with D and crosses with the \mathbf{K}^+ , $N_B=0$ and \mathbf{K}^+ , $N_B=1$ levels at quite large D . This could occur at smaller D if a small positive Δ_2 is invoked (see Extended Data Fig.~6c). After the crossing, the electrons fully occupy the spin-up and spin-down \mathbf{K}^\pm , $N_B=0$ levels at $\nu=-2$. The $-3/2$ state would correspond to a half filled NR1 level as indicated in Extended Data Fig.~6d.”

3) The last paragraph of page 6 becomes somewhat challenging to follow when the authors elucidate the effect of the displacement field using Extended Data Fig. 6b. In the main text, the authors utilize K^- , NB or K^+ , NB notation, whereas Extended Data Fig. 6b employs NR0 or NR1. Consistency in notation between the main text and Extended Data Fig. 6b would enhance clarity.

Response: We thank the reviewer for pointing out this issue. It is indeed somewhat confusing to use two types of notations. In the absence of displacement fields, the N_M and N_B symbols were used because we tried to be consistent with previous works. This decomposition into MLB and BLG parts is no longer valid when D is nonzero, so we have to use NR0/NR1 to emphasize their orbital contents. We have made a few changes to address this question.

1) In Extended Data Fig 6a, we added the “ N_M ” and “ N_B ” symbols to emphasize the connection between the two types of notations.

2) In the only paragraph of page 5, we wrote “The subscript M/B traces the MLG/BLG origin of a level and the number 0/1 indicates that it is of the NR0/NR1 type.”

3) In the last paragraph of page 6, we wrote “Rigorously speaking, the Landau levels can no longer be labeled using $N_{\{B/M\}}$ etc. However, it is convenient to track the evolution of each level with D and refer to them using the names at $D = 0$ mV/nm.”

4) The authors mention that for the $\nu = 3/2$ state, the K^+ , $N_B=0$ and K^+ , $N_B=1$ levels are partially occupied. However, according to Fig. 3f, shouldn't it be that K^+ , $N_B=0$ is fully filled and K^+ , $N_B=1$ is partially filled? Clarification on this point would be appreciated.

Response: We thank the reviewer for pointing this out. The two Landau levels (K^+ , $N_B=0$ and K^+ , $N_B=1$) are very close in energy so mixing between them is strong. It is thus safer to say that both levels are partially occupied rather than saying one of them is fully filled. Indeed, the non-perturbative mixing between these two levels is essential for generating the Pfaffian state. We have made one change to address this question.

1) In the second paragraph of page 6, we wrote “For the $\nu=3/2$ state, the levels that should be considered are $\mathbf{K}_{\{+\}}, N_{\{B\}}=0$ and $\mathbf{K}_{\{+\}}, N_{\{B\}}=1$.”

Reviewer #2 (Remarks to the Author):

This manuscript reports the observation of even-denominator (ED) fractional quantum Hall (FQH) states in ABA ‘bernal’ trilayer graphene. Even-denominator states have been of great interest for some time due to the possibility of using certain such states them to observe and control nonabelian anyons. However, not all even-denominator ground states support such phenomena. In monolayer graphene, even-denominator states that are not expected to support nonabelian anyons arise due to the roles of the spin and valley degrees of freedom and are thus referred to as ‘multicomponent’. On the other hand, in bilayer graphene, even-denominator states that so far appear to be consistent with those predicted to exhibit nonabelian anyons have been observed. These states may be referred to as ‘single-component’ in that they are predicted to be spin and valley polarized, and the origin of the even denominator is thus ‘pairing’ between Landau levels (LLs) with different indices. Given this situation, ABA trilayer graphene is an interesting system for exploring this kind of physics because its band structure consists of coexisting massless monolayer-like and massive bilayer-like bands whose hybridization can be tuned via the electric field. The authors observe four even-denominator states, of which they attribute three to the bilayer-like massive bands and one to the monolayer-like massless bands. This raises the possibility that the three states observed in the massive bands may support nonabelian anyons.

From my point of view this manuscript is suitable for publication in Nature Communications in its present form. In terms of the technical side of the experiment in my mind there is no question that everything is of high quality and that the identification of the states and related data interpretation is reliable. The observations are new, and this data set on QHE in ABA trilayer is to my knowledge the highest quality available in the literature. What is especially interesting in this work is that this is the first platform that may have the possibility of ‘tuning’ between

potentially nonabelian versus multicomponent ED states in a single device. This feature would seem to open lots of doors for future work, in particular probing the isospin order in the various ground states or studying more complex geometries. For example one could imagine making devices with layer discontinuities (e.g. at a trilayer/bilayer interface) which would presumably be useful for probing the properties of the incompressible state. Because of the new possibilities it raises I think the work is novel and notable enough to warrant publication.

Response: We thank the reviewer for very accurate summary of our manuscript and for the enthusiastic recommendation of publication.

There is one aspect I would like to ask about and that the authors could perhaps expand on in the manuscript if they think it is warranted. How well established is the level ordering depicted in Fig. 3f? It is of course difficult to directly determine the spin and especially orbital structure of various ground states, but in particular, is it really established that the claimed orbital splitting indeed exceeds the Zeeman splitting at high fields (~ 4.6 meV claimed by the authors versus ~ 1.6 meV at 14 T)? Even indirect experimental evidence would be helpful as one would like to be sure what role the spin is playing.

Response: We thank the reviewer for bringing up this issue. As the reviewer noticed, it is difficult to determine the single-particle parameters precisely. One reason is the lack of experimental tools that can directly reveal the energy levels. For bilayer graphene (BLG), asymmetric capacitance measurement have been used to extract the parameters [Nature Communications 8, 948, (2017)]. However, there is no theoretical proposal or experimental demonstration in trilayer graphene (TLG) to the best of our knowledge. Another reason is that electron correlation could be significant in multilayer graphene. For example, strongly correlated phases have been observed in BLG at zero magnetic field [Science 375, 774 (2022); Nature Physics 18, 771 (2022); Nature 608, 298 (2022)]. The Δ_2 parameter in the tight-binding model of TLG is related to the electrostatic interaction in the vertical direction. It is quite difficult to separate single-particle and interaction effects so precise determination of the parameters is challenging. Experimental and theoretical investigations in BLG suggest that Zeeman splitting is smaller than the orbital splitting [Nature Communications 8, 948, (2017); Nature 549, 360 (2017)]. In the absence of displacement fields, the Landau levels of TLG can be decomposed into monolayer and bilayer parts. While the parameters of these decomposed parts are different from the actual monolayer and bilayer systems, the mechanisms that lead to the orbital splitting are similar. Having these difficulties in mind, while we cannot be absolutely sure, it is still quite reasonable to believe that the Zeeman splitting is smaller than the orbital splitting.

Reviewer #3 (Remarks to the Author):

The paper presents experimental observations of FQHE in trilayer graphene versus magnetic field strength and vertical voltage (called as a displacement field). Though the bilayer graphene has been explored relatively widely with regard to quantum Hall behavior, the trilayer one was not a subject for such measurements in the past. Thus the experimental study of trilayer graphene is of large significance.

Response: We thank the reviewer for a careful reading of our manuscript and for concluding that “the experimental study of trilayer graphene is of large significance.”

The results are interesting but the interpretation of experimental data and theoretical comments must be revised, as are frequently confusing. The quality of colorful figures is guaranteed by plotters of experimental facilities, but the commentary part developed by Authors is insufficient and partly wrong if compared to former studies of bilayer graphene, especially in higher LLs.

To be more specific the following phrases/issues must be revised:

“And fully filling such Landau levels (LLs) one by one gives rise to the integer quantum Hall states [1]. Within one LL, the strong Coulomb interaction dominates over kinetic energy in the highly degenerated LL flatband, and further conduces to the emergence of FQH states at certain fractional fillings ν [2].”

Both these phrases are incorrect. In fact both IQHE and FQHE arise due to strong correlations of interacting 2D electrons at the magnetic field presence. The correlations are induced by the interparticle interaction in 2D geometry. Important is a cyclotron commensurability of braid trajectories with Wigner lattice of repulsing electrons deposited on positive planar jellium. The commensurability conditions are topological invariants robust against local imperfections and small temperature chaos (thus IQHE and FQHE are observable at low temperatures, even though the perfect classical Wigner lattice is defined for $T=0$ K). The patterns of the cyclotron commensurability (topological invariants) are discrete versus magnetic field value and define a general hierarchy of filling fractions embracing fillings ratios at which IQHE and FQHE manifest themselves. Correlations lead to an energy gain of multiparticle collective system of all electrons (counted with respect to noninteracting multiparticle system) and IQHE (similarly as FQHE) does not exist for non-interacting electrons (as Wigner lattice requires electron repulsion, though the interaction does not enter here in a perturbative manner). The complete fillings of LLs are possible also for gaseous systems of fermions, which do not exhibit, however, IQHE (the coincidence of the correlated wave function at $\nu=1$ with Vandermonde polynomial for non-interacting fermions is a source of frequent confusion). Thus fully filling of LLs does not give rise to the IQHE (despite a common but false popular opinion – IQHE is not a single-particle phenomenon, but is a collective topological effect of interacting electrons, similar as FQHE) – cf. [b] *Annals of Physics* 430 (2021) 168493.

Response: We thank the reviewer for providing another theoretical understanding that the IQHE is due to electron correlations. Our manuscript is an experimental work, in which we do not observe evidence that IQHE is due to electron correlations, so our experiment cannot support that theory.

Actually, the theory of IQHE, which is well-accepted by the physics community, can be found in various textbooks such as

1a) G. D. Mahan, *Condensed Matter in a Nutshell*, Princeton University Press.

1b) B. A. Bernevig and T. L. Hughes, *Topological Insulators and Topological Superconductors*, Princeton University Press.

- 1c) S. M. Girvin and K. Yang, *Modern Condensed Matter Physics*, Cambridge University Press.
1d) S. Sachdev, *Quantum Phases of Matter*, Cambridge University Press.

D. J. Thouless and F. D. M. Haldane were awarded the Nobel prize in 2016 partially for their works on the IQH effect. As far as we can see, it is the consensus of the condensed matter physics community that the IQH effect can be explained very well using non-interacting electrons in most cases.

It is possible that some IQH states are generated by electron correlation, but this does not invalidate our description. The reviewer mentioned another theory of IQH effect in which Wigner lattice plays a prominent role. However, experimental evidence for Wigner crystal has only been reported in some cases, and does not always occur concomitantly with IQH state. The reviewer mentioned “The complete fillings of LLs are possible also for gaseous systems of fermions”. Unfortunately, we are not aware of such experiments. The closest setting that we can imagine is ultracold atoms in appropriate confinement potentials, but the atoms are neutral and one can only generate effective LLs using artificial magnetic fields.

Next, It must be emphasized that the old model of composite fermions (CFs) is obsolete and incorrect in view of many new experiments. CF model fails in explanation of FQHE in so-called enigmatic states in the LLL (like 5/13, 4/11, 3/10, 3/8 and many others), in higher LLs of GaAs – cf. [a] (FQHE states at fractions with denominator 3, as those reported in the paper, are not of CF type in higher LLs), in bilayer graphene – cf. [d], in two layer GaAs systems both with [i,j,a] and without [a] carrier interlayer hopping, in monolayer graphene at $\nu=1/2, 1/4$ [g], at $1/2$ state in bilayer graphene in suspended sample [e], in hole 2DHS GaAs at $\nu=3/4$ [k] and in many others. Such an abundance of counterexamples against CF model arises from a wrong idea about effective quasiparticle with artificial fluxes attached. None fluxes are attached in fact to electrons (a hypothetical magnetic field, quanta of which are assumed to be pinned to CFs, does not exist – this field is only an auxiliary fictitious construction to mimic additional loops in 2D multiloop braids, and in addition CF model does it not correctly). It has been demonstrated [b] that CF model confuses an important parameter in the Jain hierarchy $\nu=n/((q-1)n +(-)1)$, n positive integer – the ratio of a total number of electrons in 2D system to a number of next nearest neighbors of consecutive rank in Wigner lattice with the Landau index in artificial and auxiliary spinless gaseous fermion system. Though the latter is also an integer, not all integers are in fact possible, due to the structure of next-next-nearest neighbors in Wigner lattice, which was not taken into account in CF model. The wave functions in CF model assumed as projected from higher LLs (in the spinless auxiliary LL model) onto LLL (to remove singularities) are wrong compared to the true multiparticle wave functions obligatory keeping the braid symmetry [b]. Therefore referring to CFs in the paper is confusing. Next, the Ref. [20] cited in the paper presents a naive attempt to lift the CF model to account bilayer Hall systems (in bilayers we deal with the natural distribution of loops of multiloop cyclotron braid orbits among layers, which authors of [20] try to model by strange division of flux quanta attached to electrons). Similarly, in a series of recent CF continuation papers promoted in APS journals and trying to introduce next artificial constructions to CFs like CF interaction or “partons” (probably to mimic loops in multiloop braids) is a rather incorrect interpretation favoured by lobby of supporters, which still in a biased manner protects CF model despite its evident failure in view of majority of Hall experiments and internal inconsistency in the CF model.

Response: We thank the reviewer for this detailed comment on the composite fermion (CF) theory. Based on what we have learned from the literature, the CF theory is still being actively used by various researchers to explain recent experimental results. In addition to some papers that we have cited, a few other papers are

- 2a) X. Liu et al., Nature Physics 15, 893 (2019).
- 2b) J. I. A. Li et al., Nature Physics 15, 898 (2019).
- 2c) M. S. Hossain et al., Nature Physics 17, 48 (2021).
- 2d) V. Shingla et al., Nature Physics 19, 689 (2023).
- 2e) J. Cai et al., Nature 622, 63 (2023).
- 2f) H. Park et al., Nature 622, 74 (2023).
- 2g) Z. Lu et al., Nature 626, 759 (2024).

We respectfully disagree with the reviewer's claim "the old model of composite fermions (CFs) is obsolete and incorrect in view of many new experiments." The FQH states at filling factors $5/13$, $4/11$, $3/10$, $3/8$ in GaAs have been studied in the CF framework, see for example

- 2h) S Mukherjee et al., Phys. Rev. Lett. 109, 256801 (2012).
- 2i) S. Mukherjee et al., Phys. Rev. Lett. 112, 016801 (2014).
- 2j) S. Mukherjee and S. S. Mandal, Phys. Rev. B 92, 235302 (2015).

The reviewer also mentioned a few even-denominator states in GaAs and graphene (or its bilayer). It is certainly true that even-denominator FQH states are not explained as IQH states of CFs, but this does not imply that the CF theory is wrong. Firstly, every theory has its applicable range. Secondly, some of these even-denominator FQH states may be understood as paired CF states. Of course, there are still many unclear aspects and further investigations are certainly needed.

Once again, we thank the reviewer for providing alternative explanations that do not rely on composite fermions. While we respect this academic opinion, our experimental data does not contradict existing theoretical framework. The reviewer also mentioned "a series of recent CF continuation papers promoted in APS journals ...". It is our understanding that these papers represent a consensus of the community, which we tend to agree with. If there is any debate on the theoretical side, our experimental results do not provide enough evidence to make the claim that "CF theory is incorrect".

Authors address also to $5/2$ state (which actually is of paired type also in topological approach, confirmed by exact diagonalization in toy models, on the other side) and to so-called "non-Abelian anyons". As reported in the paper the state $9/2$ may also be of paired character (but $1/2$ and $3/2$ not, what is clear in the braid approach), such a comment needs, however, a more clear explanation. "Non-Abelian anyons" are not quasiparticles, they are different notion than Abelian anyons. The latter are particles satisfying a fractional statistics – non-Abelian anyons are linked to matrix $n \times n$ ($n > 1$) unitary representations of the braid group and are related to a possible degeneracy of excitations transforming along such matrices. They are not "protected by

topological invariants”, but may serve to approximate (if are sufficiently dense) any unitary transformation of an universal gate for QIP according to Kitaev idea. Moreover, the idea of topological quantum computing cited by Authors bases on a model of an Abelian anyon on torus (but on torus scalar unitary representations of braid group – Abelian anyons – do not exist [1]). A false assumption makes all next implications logically true, which challenges to some extent the cited paper Ref. [15]. This must be clarified, how to understand non-Abelian anyons, if they are needed in the paper.

Response: We do not understand why the reviewer claims that “Non-Abelian anyons’ are not quasiparticles”. As far as we can see, it is well accepted by the condensed matter physics community that non-Abelian anyons are quasiparticles that can be defined mathematically and may appear in a few physical systems. There is a comprehensive review on this topic:

3a) C. Nayak et al., Rev. Mod. Phys. 80, 1083 (2008).

Some recent papers about non-Abelian anyons that are published on Nature Portfolio journals are:

3b) O. Tanaka et al., Nature Physics 18, 429 (2022).

3c) J.-Y. M. Lee and H.-S. Sim, Nature Communications 13, 6660 (2022).

3d) K. Hwang et al., Nature Communications 13, 323 (2022).

3e) T. I. Andersen et al., Nature 618, 264 (2023).

3f) M. Iqbal et al., Nature 626, 505 (2024).

Our observation of even-denominator FQH states whose elementary excitations are non-Abelians provide another valuable platform for studying these exotic quasiparticles. It is beyond the scope of this work to study the mathematical structure of non-Abelian anyons.

Finally Authors try to add a significance to their experimental observations addressing to the field theory (in summary) – this is also confusing a bit, as the field theory of Chern-Simons type is not fundamental – this is only a formal exemplification of artificial fluxes attached to fermions or bosons to change the statistics in 2D by hands on demand. Chern-Simons field is not a canonical transformation and despite the name of “gauge field”, the resulting shifted statistics (and auxiliary fluxes added to anyons or composite fermions) are not derived here from first rules but are by hands assumed/selected in the same artificial manner as for CF model. Over 30 years lasting story of such an approach to FQHE without any significant progress evidences that CF (and related formulation in field theory terms) is not helpful, if the homotopy background of quantum Hall physics is neglected.

Response: The Chern-Simons theory is mentioned in the last paragraph as an outlook about future directions. It is not our intention to discuss if Chern-Simons theory is fundamental or not. Topological field theory has been studied extensively and applied with considerable success in studying FQH states. Emergent gauge fields play an important role in Chern-Simons theory even without invoking CFs. If we can induce continuous phase transitions of FQH states, it is reasonable believe that the critical points are described by exotic field theory.

The paper has a chance to contribute to the domain of FQHE in unexplored as of yet region of trilayer graphene, but must take into account the current knowledge avoiding continuation of

apparently misleading obsolete ideas (despite they are widespread and indiscriminately repeated in many papers mostly experimental, but are incorrect).

If Authors want to comment on the reason of observed FQHE features at filling ratios in higher LLs in trilayer structure, two facts must be included,

1) various patterns of the distribution of loops of multiloop braids among layers in extension of the behavior known from bilayer graphene [d] (in particular fractions with the denominator 5), 2) the size of cyclotron orbits in higher LLs larger than in the LLL [a], which is important for the commensurability invariants. These two circumstances decide about the general hierarchy of homotopy patterns (filling fractions) in multilayer 2D structures including higher LLs also in trilayer graphene. Note that not all patterns from the general hierarchy are observable, and the experimentally detected hierarchy can be tuned by various factors deciding on stability of particular patterns (stability is controlled by the envelope part of the multiparticle wave function dependent on single-particle LL wave functions contaminated by a specific crystal field in graphene and in multilayer structures – here is a place for various symmetry breaking language).

Response: We thank the reviewer for the comments, there may be some misunderstanding on “filling ratios in higher LLs in trilayer structure.”. The Landau levels between -6 and 6 are generally referred to as the zeroth Landau level of trilayer graphene. The single-particle wave functions are complicated that consist of many non-relativistic Landau orbitals.

The discussion of the results needs thus a major revision. In view of the above comments the formulation of the summary (and in many other places in the text) seems to be insufficient/missed in part. Some elusive comments about various symmetry breaking as the source of complicated hierarchy structure do not have here a central significance – the complicated hierarchy of filling ratios is a matter of homotopy invariants. The observed stability of particular homotopy patterns changing in response to magnetic field value (at the same filling ratio as in a fan diagram) or to vertical electric field, does not concern the general hierarchy but only competition between various patterns available at same filling fractions (which is explained within topological homotopy approach [b], but not in CF model). This has been demonstrated earlier experimentally, like for the stability of FQHE at $1/2$ or $1/4$ in monolayer graphene at some window for magnetic field [g,f], stability of $1/2$ FQHE state in bilayer graphene [in suspended sample [e] though not in a sample deposited on hBN substrate), stability of FQHE at $\nu=3/4$ in 2DHS in GaAs versus Hall metal at $\nu=3/4$ in 2DES in GaAs at similar other conditions [k], at the demonstrated transition of FQHE bilayer hierarchy to monolayer one by application of vertical electric field [h], cf. also SI to [d] for explanation. All these have been explained in homotopy terms, apparently beyond the CF model, which is unable to consider such situations.

The same here – in the paper Authors use the improper theoretical methodology (in fact only phraseology repeated from many former publications, where similar experiments were reported without, however, successful explanations if neglected the braid homotopy conditioning of FQHE hierarchy). In the case when the assumptions of CF model are fully clarified (including its limitations [b]), CF model approach does not conform with the already gained knowledge and the paper must be revised to avoid confusions. The addressing to bizarre concepts of CF theory

developments as an interaction of CFs or partons is similar to producing epicycles to epicycles to fit reality of completely different character.

For example, some phrases on filling fraction denominators can be highly deceptive if addressed to higher LLs in multilayer structure (e.g., as shown in [a] the denominators 3 are not obligatory referred to three-loop orbit, unless in the LLL in GaAs; in higher LLs in GaAs [a] they are related to single-loop orbit precluding any CF – three loop – interpretation). In graphene the situation is even more complicated because of spin-valley subband structure and odd denominators [d] may not be related to a simple case in the LLL of GaAs, where CF intuitions might be accepted (though not for even denominator fractions or other enigmatic states).

Response: We thank the reviewer for providing alternative explanations. Despite our best effort, we are not able to see how the topological homotopy approach can explain our experimental observations. The reviewer mentioned a few other even-denominator FQH states, but a proper investigation of their nature is beyond the scope of our manuscript. Any debate on this subject should rather be presented in separate papers that focused on theoretical aspects.

Minor comments – lettering in Bibliography should be revised. In the sentence “A higher magnetic field is require to do the same thing for the K– valley.” “required” should be rather. In caption to Fig.1 “The filling factors defined at B = 14 T is given below the bottom axis.” It should be rather “are”.

Response: We thank the reviewer for spotting these typos and they have been corrected. The manuscript has been carefully proofread and some other errors are also corrected.

mentioned above references.

a. J. Jacak and L. Jacak, The commensurability condition and fractional quantum Hall effect hierarchy in higher Landau levels. *JETP Letters* 102, 19 (2015).

b. J. Jacak, Topological approach to electron correlations at fractional quantum Hall effect, *Annals of Physics* 430, 168493 (2021).

c. J. Jacak, Superfluidity of indirect excitons vs quantum Hall correlation in double Hall systems: Different types of physical mechanisms of correlation organization in Hall bilayers, *Phys. Lett. A* 382, 41 (2018).

d. J. Jacak, Unconventional fractional quantum Hall effect in bilayer graphene, *Scientific Reports*, 7, 8720 (2017).

e. D. K. Ki, V. I. Falko, D. A. Abanin, A. Morpurgo, Observation of even denominator fractional quantum Hall effect in suspended bilayer graphene, *Nano Lett.* 14, 2135 (2014).

f. J. Jacak, Explanation of an unexpected occurrence of $\nu = \pm 1/2$ fractional quantum Hall effect states in monolayer graphene, *J. Phys.: Condens. Matter* 31, 475601 (2019).

- g. A. A. Zibrov, E. M. Spanton, H. Zhou, C. Kometter, T. Taniguchi, K. Watanabe and A. F. Young, Even denominator fractional quantum Hall states at an isospin transition in monolayer graphene, Nat. Phys. 14, 930 (2018).**
- h. P. Maher, L. Wang, Y. Gao, C. Forsythe, T. Taniguchi, K. Watanabe, D. Abinin, A. Papic, P. Cadden-Zimansky, J. Hone, P. Kim and C. R. Dean Tunable fractional quantum Hall phases in bilayer graphene. Science 345, 61 (2014).**
- i. Y. W. Suen, L. W. Engel, M. B. Santos, M. Shayegan, D. C. Tsui, Observation of a $\nu = 1/2$ fractional quantum Hall state in a double-layer electron system, Phys. Rev. Lett. 68, 1379 (1992).**
- j. P. Eisenstein, G. S. Boebinger, L. N. Pfeiffer, K. W. West, S. He, New fractional quantum Hall state in double-layer two-dimensional electron systems, Phys. Rev. Lett. 68, 1383 (1992).**
- k. C. Wang, A. Gupta, S. K. Singh, Y. J. Chung, L. N. Pfeiffer, K. W. West, K. W. Baldwin, R. Winkler, M. Shayegan. Even-denominator fractional quantum Hall state at filling factor $\nu = 3/4$. Phys. Rev. Lett., 129, 156801 (2022)**
- l. T. Einarsson, Fractional statistics on a torus, Phys. Rev. Lett. 64, 1995 (1990)**

REVIEWERS' COMMENTS

Reviewer #1 (Remarks to the Author):

I find the changes that the authors made satisfy the concerns I raise in my review. I now believe that this work can be published in Nature Communications.

Reviewer #2 (Remarks to the Author):

In my opinion the authors have more than adequately addressed the questions that I raised, as well as those of the other reviewers. I recommend publication.